# Prevalence of falls and comparison of health-related physical fitness factors between different faller categories among institutionalized older adults in Kandy District of Sri Lanka

**Welgama Ihalage Suheja Madhushani Ihalage**[ID]\*, **Vidhana Ralalage Chalana Sithara Wijebandara, Diwalawaththe Gedara Wathsala Sewwandi Wickramakumari, Wickramasinghe Mudiyanselage Buddhini Dilesha Wickramasinghe, Rathnayaka Mudiyanselage Ruwan Keerthi Sampath, Manchanayake Mudiyanselage Jinali Pabodha Manchanayake, Esther Liyanage**

Faculty of Allied Health Sciences, Department of Physiotherapy, University of Peradeniya, Peradeniya, Sri Lanka

\* suheja2017@gmail.com

## Abstract

Falls can be identified as one of the main issues in elderly population which can lead to serious consequences. Elderly population can be subdivided as community dwelling older adults and institutionalized older adults. The components of health-related physical fitness factors play an important role in the perspective of healthy ageing. The objective of the study was to determine the prevalence of falls and to compare health-related physical fitness factors between different fall categories among institutionalized older adults. This descriptive cross-sectional study comprised of one hundred and seventy-two elders above 60 years of age, living in registered elder's homes in Kandy District. The prevalence of falls was determined by the number of falls reported by the subjects. They were further categorized as non-fallers, fallers, and frequent fallers depending on the number of falls. Body Mass Index (BMI) for body composition, 2-minute walk test for cardiovascular endurance, 30 second sit to stand test for muscle strength and endurance, hand grip strength for upper body strength, chair sit and reach test for lower body flexibility and back scratch test for upper body flexibility were the measures used to assess health-related physical fitness factors. The prevalence of falls is presented as percentage and the health-related factors were compared between the three faller categories using one-way ANOVA and pairwise comparison was performed using Scheffe test. The prevalence of falls was 47.1%. There was a significant difference in BMI, cardiovascular endurance, and lower body flexibility between the three categories of fallers. Higher BMI, lower cardiovascular endurance and lower flexibility in the lower body were associated with increased prevalence of falls (p<0.05). There was no significant difference in body fat percentage, hand grip strength, lower body muscle strength and endurance and upper body flexibility (p>0.05). The findings suggest that, BMI,

**Data Availability Statement:** All relevant data are within the paper and its Supporting Information files.

**Funding:** No funding was received for this work.

**Competing interests:** authors declare no competing interest.

cardiovascular endurance, and lower body flexibility must be addressed and managed, while designing intervention programs for falls prevention among institutionalized older adults.

## Introduction

The life expectancy of humans all over the world has increased. Currently, most people reach their sixties and beyond. Both the size and the proportion of elderly in the population have shown a growth in every country [1]. It is anticipated that by 2030, 1 in 6 individuals within the population will be 60 years of age or beyond. One billion people aged 60 years, and above which was calculated in 2020 is expected to be 1.4 billion by 2030. By, 2050 this population will be doubled (2.1 billion). A threefold increment is expected in the population aged 80 years or older between 2020 and 2050 (426 million) [1].

The shift in the population distribution of a country towards elderly is known as the "population ageing". Population ageing initially started in high-income countries, as an example the population aged over 60 years is 30% in Japan. But now the greatest change is observed in low- and middle-income countries. Two-third of world's population over 60 years is expected to be in low- and middle-income countries by 2050 [1]. When considering South Asian countries, Sri Lanka was the first to achieve demographic transition. In 2012, Sri Lanka had higher proportions of elderly population compared to other South Asian countries. The ageing process in Sri Lanka is slightly more rapid than other countries in South Asia since 1980s. Percentage of population over 60 years and beyond almost doubled from 6.6% in 1981 to 12.4% in 2012. Again, it is expected to be doubled up to 24.8% by 2041. By 2041, 1 in 4 persons in Sri Lankan population will be an elder. But other countries in South Asia are expected to have lower rates [2].

According to the current available statistics, 12.3% of the total population in Sri Lanka is aged 60 years or above [3]. When considering the elderly population in a country, it can be mainly subdivided as; community dwelling older adults and, institutionalized older adults. The recent changes in the economy as well as the society have paved the way for an increasing proportion of older adults spending their lives in elderly care institutions [4]. It was only 1% of the total elderly population in Sri Lanka that was institutionalized in 2012 [5].

There are significant differences observed between community-dwelling and institutionalized older adults. Quality of life (QOL) of community-dwelling older adults has been reported to be a moderate level while the institutionalized older adults have a relatively poor QOL [6]. Quality of life has shown to be related to falls. Findings from a study have found that the elders with impaired quality of life in physical aspect, physical and emotional capacity, reported to have increased number of falls [7]. Limitations for social and physical activities are also higher for institutionalized older adults [8]. Among health-related physical fitness factors lower body strength and cardiovascular endurance have shown to be higher among community dwelling older adults in India [9]. Evidence from literature shows that, institutionalization has brought about various negative impacts to the lifestyle of geriatric population in Sri Lanka.

A fall can be defined as an event which results in a person coming to rest inadvertently on the ground or floor or other lower level [10]. When considering falls, comprehensive evaluation of falls and recurrent falls is important. A faller can be defined as an individual who had at least one fall in past 12 months. And a recurrent faller can be defined as an individual who had

2 or more than 2 falls in the past 12 months [11]. In a report by the WHO in 2021, it was stated that over 80% of fall-associated fatalities occur in low- and middle-income countries. The increase in risk of falls with ageing also increases the risk of death or serious injury arising from a fall among older adults [10].

There is sufficient evidence that the proportion of ageing population is increasing rapidly world-wide. However, along with this increase there is a challenge to maintain good health, safety, and performance among these ageing people. Falls is identified as a major risk for injuries and death among the older adults. There are various factors that increase fall risk and each of the health-related physical fitness parameters are few among those factors that have shown to decline with ageing and are essential factors to prevent falls [12–16], maintain health and improve performance. A thorough literature search revealed that, there is dearth in published information about the association between health-related physical fitness parameters and falls. In addition, there was no published study that assessed this association among institutionalized older adults in Elderly Homes in Sri Lanka. There are 12 elders' homes in Kandy district which are registered under the Department of Social Welfare, Probation and Child Care Services namely: Udunuwara Elders' home with 16 elders, Mahaiyawa Elders home with 47 elders, Mulgampola Elders' home with 69 elders, Pihibiyahena Elders' home 17 elders, Mawilmada Saranasevana Elders' home with 23 elders, Godahena Elders' home 22 elders, Ampitiya Elders' home with 20 elders, Nikaketiya Elders' home with 13 elders, Shamrock Elders' home with 17 elders, Bopitiya elders' home with 23 elders, Katugasthota Sri Subodhi Elders' home with 20 elders and Kotikamba Senior Citizens' home with 17 elders. There were 304 elders in total in the twelve Elders' homes. The aim of the present study was to determine the prevalence of falls and to compare health-related physical fitness factors: body composition, cardiovascular endurance, lower body strength and endurance, upper body strength, lower body flexibility, upper body flexibility between non-fallers, fallers and frequent fallers among the older adults in Elderly Homes in Kandy District.

## Materials and methods

### Study samples and procedures

The study was a descriptive cross-sectional study design. Institutionalized older adults 60 years and above living in Elderly Homes in Kandy District registered under Department of Social Welfare, Probation and Child Care Service were the study sample. The study excluded older adults living in an Elderly Home for less than one year, a Short Blessed Test (SBT) score greater than 4, major psychiatric illnesses, depression, malignancies, severe hearing or visual impairments, neurological illnesses, bedridden and severely ill.

The total number of elders was known to be 304, therefore, the sample size was calculated using Slovin's formula [17]. According to the formula where "N" was the total population and "e" was margin of error (5%), a sample of 172 was obtained. Considering a dropout rate of 10%, the total sample size for the study was 190. Cluster Random Sampling was used. According to this method, the names of the Elderly Homes in Kandy District registered under Department of Social Welfare, Probation and Child Care Service were written and placed in sealed envelopes. One envelope was picked at a time randomly and the name of the Elderly Home and the number of samples available was recorded. This procedure was continued until the number of samples in the Elderly Homes reached the number required by sample size calculation. Accordingly, Udunuwara Elders' home, Mahaiyawa Elders home, Mulgampola Elders' home, Pihibiyahena Elders' home, Mawilmada Saranasevana Elders' home, Godahena Elders' home, and Ampitiya Elders' home were selected as the study setting. Out of 214 older adults, 42 of them were excluded because of not satisfying the inclusion criteria.

## Measures

Socio-demographic data comprised the age, gender, handedness, information regarding comorbidities, medication, and history of surgeries was obtained using interviewer-administered questions. Prevalence of falls was determined by the reported number of falls sustained during the past one-year. The subjects were categorized as non-fallers, fallers, or frequent fallers according to the number of falls. The subjects who did not experience any falls during the past year were categorized as non-fallers, those who sustained a single fall were categorized as fallers and those with a history of two or more falls in the past year were categorized as frequent fallers [11]. Different physical fitness tests were used to assess each component of health-related physical fitness; Body Mass Index (BMI) and bioelectrical impedance analysis for body composition, 2 minute walk test for cardiovascular endurance, 30 second sit to stand test for lower body strength and endurance, hand grip strength measure for upper body strength, chair sit and reach test for lower body flexibility, and back scratch test for upper body flexibility. To determine sufficient memory to recall number of falls among the subjects, SBT was used. As this tool was available in English language, it was translated to Sinhala and Tamil languages and pre-tested before use in the study. All the questions were administered by interviewer-administered method. Written informed consent was obtained from the subjects before participating in the study.

**Tests of measurement.  Body composition** can be estimated by Body Mass Index (BMI) and body fat percentage. While the participant was barefooted and stood straight, the height was measured using a stadiometer. BMI and body fat percentage was evaluated by Bioelectrical impedance analyzer. To measure body composition using Karada Scanner, first the personal information of the subject including age, gender, and height were entered to the machine. Heels of the subject were positioned on heel electrodes so that the weight of the subject was evenly distributed on the measurement platform. The subject remained in the same position and the weight measurement was taken. After the screen displayed "START" command, the subject was guided to lift the display unit off the main unit, and to hold it in front of the body. While maintaining this position, the subject's knees and back were kept straight, shoulders 90 degrees flexed, and elbows extended. After the measurement was completed, the subject stepped off the main unit. Measurements taken were displayed on the display unit.

## 2-Minute walk test

A long flat corridor was selected to perform the test. A walking area of 15 m was selected and every 1m distance was marked using colored strips. Two traffic cones were placed at either end. Subjects wore comfortable light clothing. Subjects who were using walking aids were allowed to use it during the test. Vigorous activities were not allowed 2 hours before the test. Heart rate, blood pressure and oxygen saturation were measured before and after performing the test. Subjects were advised to stop the test if there was any discomfort such as chest pain, dizziness, or undue fatigue. The subject was guided to walk at their own usual pace for 2 minutes. The test was performed only once on the same individual.

## 30 Second sit to stand test

Both lower body muscle strength and endurance were assessed using 30 second sit to stand test. A chair of height 45.7cm with a standard armrest was used for the test. The chair was placed against a wall to prevent moving. Before starting the test, the subject was seated on the chair with an erect back, and feet kept shoulder width apart. They were asked to place both hands on the armrest without keeping them across the chest. With the signal from the examiner the subject was asked to stand up from the chair followed by returning to the initial seated

position. The number of full stands completed within 30 seconds was recorded. This test was performed twice for one individual. A rest period of 1 minute between the two trials was given.

### Hand grip strength

Measurements were taken using handgrip dynamometer on both right and left sides. The subject was asked to be in seated position with the testing shoulder adducted, elbow flexed to 90 degrees, and forearm and wrist in neutral position. The researcher placed the dynamometer in the subject's hand and asked the subject to squeeze it as hard as possible. Extending the wrist during grip was not restricted. Two trials were given to one subject. A rest period of 1 minute between two trials was given.

### Chair sit and reach test

The subject was asked to be seated at the edge of the chair. The chair was placed against a wall to prevent moving. Subject's both legs were allowed to extend at knees. The subject was then asked to reach towards his/her toes and the number of inches between the extended fingers and the toes were measured. The test was performed twice on the same individual. A rest period of 1 minute between the two trials was given.

### Back scratch test back

The dominant side was tested to measure the upper body flexibility. To start the test, the subject was allowed to be in standing position, placing one arm at the lower back, moving it up the spine towards their head. The opposite arm was kept behind the neck, moving it down the spine, aiming to place the middle finger of each hand near to the other hand as much as possible. The gap between fingertips of the middle fingers of both hands was measured. The measurement could be either positive or negative. The test was performed twice on the same individual. A rest period of 1 minute between the two trials was given.

### Analysis

Data was analyzed using SPSS software (version 25). Demographic information was presented using descriptive statistics as mean and standard deviation. Shapiro-Wilk test was used to determine the normality distribution of data. The prevalence of falls was presented as percentage. One-way ANOVA test was used to compare health-related physical fitness factors between non-fallers, fallers, and frequent fallers. Further, Scheffe Test was conducted for pairwise comparison of the factors that showed statistically significant association in One-way ANOVA analysis.

### Ethical consideration

Ethical clearance was obtained from the Ethics Review Committee of Faculty of Allied Health Sciences University of Peradeniya (AHS/ERC/2022/044). Permission to conduct the study was obtained from the Department of Social Welfare, Probation and Child Care Services in Kandy District and District Secretariat- Kandy. The information about all the subjects was kept confidential throughout the study. In any event the individual information of any subject was and will not be published or revealed. The subjects were not left unsupervised at any time during data collection, and the researchers were continually with the subjects throughout the process of data collection. The data collection was carried out from 10[th] of May 2022 to 03[rd] of June 2022.

## Results

A total of 172 elderly living in registered Elders' Homes in Kandy District were selected for the study. The mean age of the subjects was $73.65 \pm 7.01$ years, the mean height was $147.76 \pm 8.64$ cm, and the mean weight was $51.26 \pm 11.54$ kg. Details of gender and the number of subjects from each Elders' Home is given in Table 1.

### Prevalence of falls

**Details of percentage of non-fallers, fallers, and frequent fallers.** When considering the study sample, 47.1% stated to have experienced a fall at least once during past 12 months. Out of them 18.6% reported frequent falls, while 28.5% stated to have sustained only one fall during the past year. This result has been depicted in Fig 1.

### Comparison of health-related physical fitness factors between non-fallers, fallers and frequent fallers

The health -related physical fitness factors were compared between the three categories of fallers using one-way ANOVA test. The results of the analysis are presented in Table 2.

The results in Table 2 show that there was a significant difference in BMI between the fallers, non-fallers, and frequent fallers ($p<0.05$). Cardiovascular endurance was significantly different between the three groups ($p<0.01$). Also, there was statistically significant difference in lower body flexibility (($p<0.01$). There was no statistically significant difference between the three groups of fallers for body fat percentage, lower body strength and endurance, upper body strength, and upper body flexibility ($p>0.05$), Further, Scheffe Test was conducted for pairwise comparison of BMI, cardiovascular endurance and lower body flexibility that showed statistically significant difference between the three groups of fallers.

### Pairwise comparison of BMI, cardiovascular endurance and lower body flexibility between non-fallers, fallers, and frequent fallers

The results of Scheffe test for pairwise comparison of BMI, cardiovascular endurance and lower body flexibility are presented in Table 3.

The results in Table 3 reveal there was no statistically significant difference in BMI between non-fallers and frequent fallers ($p>0.05$), fallers, and frequent fallers ($p>0.05$) or among non-fallers and fallers ($p>0.05$). When considering the cardiovascular endurance, a significant difference was seen between the groups: non-fallers and frequent fallers ($p<0.05$) and between fallers and frequent fallers ($p<0.05$). The mean values were significantly lower among frequent fallers when compared to non-fallers (-13.99) and fallers (-14.80). When assessing the results

**Table 1. Details of the participant from each Elders' Home.**

| Name of the Elders' Home | Male | Female | Total |
|---|---|---|---|
| Udunuwara Elders' home | 5 | 8 | 13 |
| Mahaiyawa Elders home | 12 | 26 | 38 |
| Mulgampola Elders' home | 18 | 46 | 64 |
| Pihibiyahena Elders' home | 3 | 8 | 11 |
| Mawilmada Saranasevana Elders' home | 5 | 10 | 15 |
| Godahena Elders' home | - | 14 | 14 |
| Ampitiya Elders' home | 8 | 9 | 17 |
| Total | 51 | 121 | 172 |

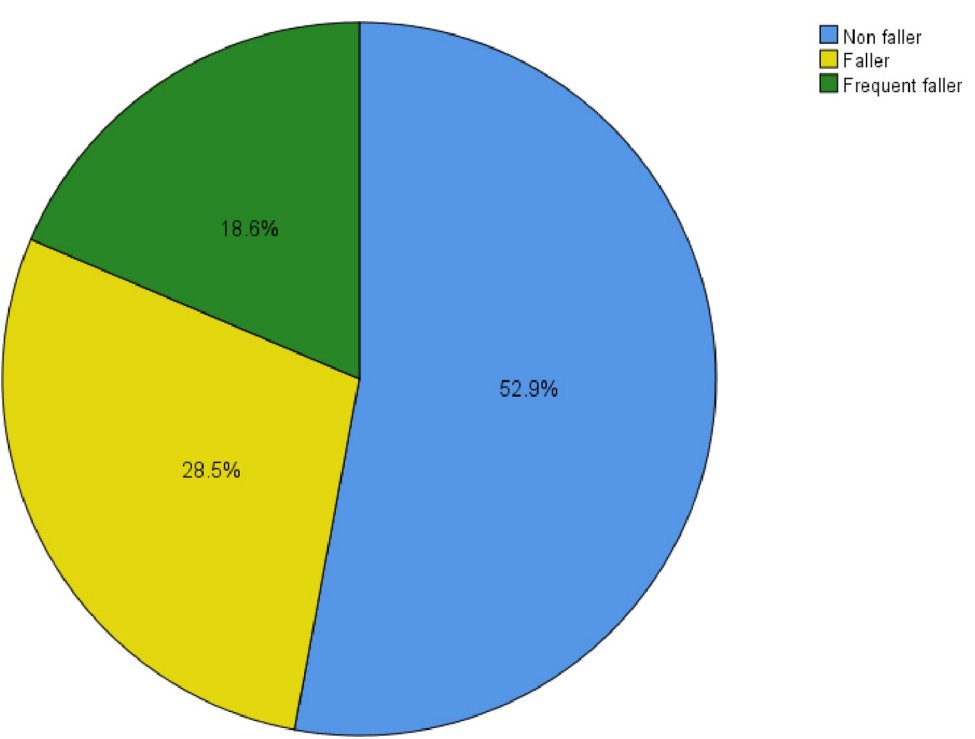

**Fig 1. Percentages of non-fallers, fallers and frequent fallers.**

for lower body flexibility, a significant difference was observed between non-fallers and frequent fallers (p<0.05). The mean values for lower body flexibility were significantly lower among frequent fallers when compared to non-fallers (-5.69).

## Discussion

The prevalence of falls among institutionalized older adults in Kandy district, Sri Lanka over a period of past 12 months was 47.1% (28.5% fallers, 18.6% frequent fallers). These findings revealed that nearly half of the study sample had experienced a fall during past year. This is a considerable proportion and implies need for attention and strategy to address the concern. Falls are avoidable incidents among older adults. To develop effective preventive strategies, the relevant causative factors should be addressed.

**Table 2. Comparison of health-related physical fitness factors between non-fallers, fallers, frequent fallers.**

| Variable | Non-faller | Faller | Frequent faller | P value |
|---|---|---|---|---|
| BMI | 22.86±4.69 | 23.26±4.34 | 32.21±42.53 | 0.044* |
| Body fat percentage | 37.15±6.87 | 35.29±7.52 | 37.26±6.85 | 0.288 |
| 2 min walk test | 71.75±24.33 | 72.55±23.78 | 57.75±20.74 | 0.009* |
| 30 sec sit and stand test | 10.55±3.76 | 10.61±2.99 | 9.03±2.65 | 0.068 |
| Grip strength (Right) | 14.51±5.66 | 14.64±5.95 | 13.97±5.97 | 0.866 |
| Grip strength (Left) | 14.20±6.17 | 14.08±4.77 | 14.10±5.77 | 0.991 |
| Chair sit and reach test | -5.64±8.17 | -7.89±8.68 | -11.33±11.37 | 0.009* |
| Back scratch test | -17.78±15.94 | -18.90±14.82 | -16.70±11.92 | 0.808 |

*p-value <0.05.

**Table 3. Pairwise comparison of BMI, cardiovascular endurance, and lower body flexibility factors between non-fallers, fallers, frequent fallers.**

| Variable | Category | | Mean difference | P-value |
|---|---|---|---|---|
| | No of falls | No of falls | | |
| BMI (kg/m$^2$) | Non-faller | Faller | -0.41 | 0.992 |
| | | Frequent faller | -9.35 | 0.054 |
| | Faller | Frequent faller | -8.94 | 0.112 |
| 2 min walk test (m) | Non-faller | Faller | -0.80 | 0.982 |
| | | Frequent faller | 13.99 | 0.017* |
| | Faller | Frequent faller | 14.80 | 0.024* |
| Chair sit and reach test (cm) | Non-faller | Faller | -2.25 | 0.37 |
| | | Frequent faller | -5.69 | 0.010* |
| | Faller | Frequent faller | -3.43 | 0.246 |

*p-value <0.05.

The prevalence of falls among institutionalized older adults who were aged 65 years or more in Portugal was 41.6% [18]. Elderly who are aged 60 years or above living in nursing homes in Malaysia had a falls prevalence of 32.8% [19]. Inconsistencies among these findings could be due to variations in facilities available at elderly care homes, culture, level of care provided to older adults, and health care facilities across different regions and countries in the world.

Few published studies have assessed the prevalence of falls and associated factors among community-dwelling older adults resided in different regions in Sri Lanka. 25.8% of community dwelling older people in Colombo district had experienced falls during the preceding year from 2010 August [20]. In Galle district, the prevalence was 34.3% in 2019 [21]. Higher prevalence of depression [4], limitations to social and physical activities [8], poor quality of life [6] and under nutrition [22] among institutionalized older adults could be a few reasons for this difference.

Results for one-way ANOVA for comparison of health-related physical fitness factors between non-fallers, fallers and frequent fallers revealed that there was a significant difference in BMI, Cardiovascular endurance, and Lower body flexibility between the three categories of fallers.

The present study concluded that a higher BMI is positively associated with falls prevalence. Similar results have been observed in a study done in Republic of Korea [23]. A mini review done in Malaysia stated that older adults with a higher BMI were more prone to falls due to poor postural balance associated with obesity [24]. An Indian study which investigated the association between anthropometric factors and balance among older adults aged 60 years and above stated that increased body fat mass and BMI was adversely associated with balance, thereby with prevalence of falls [25]. Ambulatory stumbling which is more common among obese individuals compared to healthy individuals may contribute to poor balance, thereby increased prevalence of falls [26]. Older adults with an increased BMI have a poor postural stability thus causing increased sway in comparison with older adults having a normal BMI [27]. These may be possible reasons for the association identified in the present study.

The frequent fallers had a significantly lower cardiovascular endurance in comparison to the non-fallers and frequent fallers. This finding revealed that a declined cardiovascular endurance increases fall prevalence. Cardiovascular endurance acts as a key predictor of risk of falls among older adults [28,29]. Ageing results in a decrease in cardiorespiratory endurance which in turn increases the risk of various disease conditions as well as a decline in independent

functioning of older individuals [30–33]. It is evident that maintaining cardiorespiratory fitness in older age is a key to better health through preventing the risk of major life-threatening illnesses, improving functional capacity and preservation of independence in Activity of Daily Living (ADL) [12].

Frequent fallers had significantly reduced lower body flexibility compared to non-fallers hence; it is associated with increased prevalence of falls. This result is similar to that of Toraman and Yildirim (2010); they concluded that there was a positive association between decreased lower body flexibility, mainly in the hip and ankle musculature and the prevalence of falls [34]. Lower body flexibility acts as a crucial factor in falls prevention [35].

Most published studies among elderly to assess prevalence of falls and one or more health-related physical fitness factors were among community dwelling older adults. It is believed that the findings of the present study add potentially beneficial information about prevalence of falls and its association with health-related physical fitness factors among institutionalized older adults. The present study was conducted among institutionalized older adults, there is dearth in information, rather no published research in Sri Lanka among institutionalized older adult. Hence, the findings of the study are unique and are beneficial for future studies. Also, all the health-related fitness factors were assessed among the study sample. A limitation of the study was that the prevalence of falls was determined from subjective information. However, efforts were made to minimize adverse influence by determining adequate memory of the participants and the number of falls mentioned by the participants was verified by the Elderly home staff members.

## Conclusion and recommendation

The present study concluded that the prevalence of falls among institutionalized older adults in Kandy District was 47.1%, among these, 28.5% were fallers and 18.6% were frequent fallers. There was a significant difference in BMI, cardiovascular endurance, and lower body flexibility between the three categories of fallers among institutionalized older adults. The findings suggest that improving lower body flexibility and cardiovascular endurance may prevent falls among older adults. Further, exercise programs aimed at preventing falls may address cardiovascular endurance and lower body flexibility. Results obtained in the present study can be used as baseline information to design exercise program. Further longitudinal studies to assess the health-related physical fitness factors among institutionalized older adults and interventional studies are recommended.

## Supporting information

**S1 Data.**
(XLSX)

## Acknowledgments

We are grateful to all the participants who participated in the study.

## Author Contributions

**Conceptualization:** Manchanayake Mudiyanselage Jinali Pabodha Manchanayake, Esther Liyanage.

**Data curation:** Welgama Ihalage Suheja Madhushani Ihalage, Vidhana Ralalage Chalana Sithara Wijebandara, Diwalawaththe Gedara Wathsala Sewwandi Wickramakumari,

Wickramasinghe Mudiyanselage Buddhini Dilesha Wickramasingha, Rathnayaka Mudiyanselage Ruwan Keerthi Sampath, Manchanayake Mudiyanselage Jinali Pabodha Manchanayake, Esther Liyanage.

**Formal analysis:** Welgama Ihalage Suheja Madhushani Ihalage, Vidhana Ralalage Chalana Sithara Wijebandara, Diwalawaththe Gedara Wathsala Sewwandi Wickramakumari, Wickramasingha Mudiyanselage Buddhini Dilesha Wickramasingha, Rathnayaka Mudiyanselage Ruwan Keerthi Sampath.

**Methodology:** Welgama Ihalage Suheja Madhushani Ihalage, Vidhana Ralalage Chalana Sithara Wijebandara, Diwalawaththe Gedara Wathsala Sewwandi Wickramakumari, Wickramasingha Mudiyanselage Buddhini Dilesha Wickramasingha, Rathnayaka Mudiyanselage Ruwan Keerthi Sampath.

**Supervision:** Manchanayake Mudiyanselage Jinali Pabodha Manchanayake, Esther Liyanage.

**Writing – original draft:** Welgama Ihalage Suheja Madhushani Ihalage, Vidhana Ralalage Chalana Sithara Wijebandara, Diwalawaththe Gedara Wathsala Sewwandi Wickramakumari, Wickramasingha Mudiyanselage Buddhini Dilesha Wickramasingha, Rathnayaka Mudiyanselage Ruwan Keerthi Sampath.

**Writing – review & editing:** Manchanayake Mudiyanselage Jinali Pabodha Manchanayake, Esther Liyanage.

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
