## [Decision Letter · Decision Letter 0]

16 Nov 2023

PONE-D-23-30015Prevalence of falls and comparison of health-related physical fitness factors between different faller categories among institutionalized older adults in Kandy District of Sri LankaPLOS ONE

Dear Dr. Ihalage,

Thank you for submitting your manuscript to PLOS ONE. After careful consideration, we feel that it has merit but does not fully meet PLOS ONE’s publication criteria as it currently stands. Therefore, we invite you to submit a revised version of the manuscript that addresses the points raised during the review process.

We look forward to receiving your revised manuscript.

Kind regards,

Antimo Moretti

Academic Editor

PLOS ONE

Journal Requirements:

3. PLOS requires an ORCID iD for the corresponding author in Editorial Manager on papers submitted after December 6th, 2016. Please ensure that you have an ORCID iD and that it is validated in Editorial Manager. To do this, go to ‘Update my Information’ (in the upper left-hand corner of the main menu), and click on the Fetch/Validate link next to the ORCID field. This will take you to the ORCID site and allow you to create a new iD or authenticate a pre-existing iD in Editorial Manager. Please see the following video for instructions on linking an ORCID iD to your Editorial Manager account: https://www.youtube.com/watch?v=_xcclfuvtxQ.

Reviewers' comments:

Reviewer's Responses to Questions

**Comments to the Author**

1. Is the manuscript technically sound, and do the data support the conclusions?

Reviewer #1: Partly

Reviewer #2: Yes

2. Has the statistical analysis been performed appropriately and rigorously? 

Reviewer #1: Yes

Reviewer #2: Yes

3. Have the authors made all data underlying the findings in their manuscript fully available?

Reviewer #1: Yes

Reviewer #2: Yes

4. Is the manuscript presented in an intelligible fashion and written in standard English?

Reviewer #1: Yes

Reviewer #2: Yes

5. Review Comments to the Author

Reviewer #1: I congratulate the authors for addressing an interesting and useful topic for the scientific community. in my review I will have to point out some shortcomings:

1)

The introduction talks about quality of life, depression and other aspects of the institutionalized elderly. The concept is far from the focus of the title and is not related to the risk of falls. I recommend learning more about the concepts.

2) describe the 12 homes for the elderly and the enlistment in the materials and methods but also in the introduction. I advise you to mention it only in the materials and methods.

3) You describe the names of 8 homes for the elderly, but in the results I only find 7.

4) This statement in the results should be more precise and describe only the scales "The results in Table 2 revealed that there was a significant difference in BMI, cardiovascular

lower body strength and flexibility across the three fall categories. Further,

The Scheffe test was conducted for pairwise comparison of the factors shown statistically

significant difference". Better to make the comments in discussion.

5) Table 3 is repetitive. the categories are repeated several times.

6) in the discussion the statement that institutionalization correlates to falls is incorrect as there is no group of non-institutionalized elderly comparisons

Reviewer #2: Dear Authors,

in this descriptive cross-sectional study, you aimed to determine the prevalence of falls and to compare health-related physical fitness factors between different fall categories among institutionalized older adults. The paper is well written, nevertheless there are some minor issues to be addressed:

Title: well done.

Abstract: see methods’ comment, I would do the same changes about measures description.

Introduction: I would had that there have been different study investigating the correlation between different diet/ integration and the risk of falls (for example I would suggest to cite this study DOI: 10.1007/s40520-021-01977-x).

Methods:

Measures: in this section I would be more specific describing which of the variable (bioelectrical impedance analysis, 2 minute walk test, 30 second sit to stand test, hand grip strength measure, chair sit and reach test, and back scratch test) is the measure of each outcome (Body composition, cardiovascular endurance, lower body strength and endurance, upper body strength, lower body flexibility, upper body flexibility).

Results, discussion, conclusion: well described.

6. PLOS authors have the option to publish the peer review history of their article (what does this mean?). If published, this will include your full peer review and any attached files.

Reviewer #1: No

Reviewer #2: No

---

## [Author Response · Author response to Decision Letter 0]

12 Jan 2024

Academic Editor

1. Please ensure that your manuscript meets PLOS ONE's style requirements, including those for file naming. -PLOS ONE’s style requirements were followed.

2. We note that you have indicated that data from this study are available upon request. PLOS only allows data to be available upon request if there are legal or ethical restrictions on sharing data publicly. -The data sheet of the study will be submitted along with the revised manuscript.

3. PLOS requires an ORCID iD for the corresponding author in Editorial Manager on papers submitted after December 6th, 2016. -The ORCID ID of the corresponding author is updated in the Editorial Manager Account.

Reviewer 01

1) The introduction talks about quality of life, depression and other aspects of the institutionalized elderly. The concept is far from the focus of the title and is not related to the risk of falls. I recommend learning more about the concepts. - The statement regarding sleep disorders is removed from the introduction. 

- Information about quality of life and risk of falls in included.

2) Describe the 12 homes for the elderly and the enlistment in the materials and methods but also in the introduction. I advise you to mention it only in the materials and methods. Addressed

3) You describe the names of 8 homes for the elderly, but in the results I only find 7. The stated number of Elders’ homes in Material and Methods is seven and it is in line with (same as) the number of elderly homes stated in results.

4) This statement in the results should be more precise and describe only the scales "The results in Table 2 revealed that there was a significant difference in BMI, cardiovascular

lower body strength and flexibility across the three fall categories. Further,

The Scheffe test was conducted for pairwise comparison of the factors shown statistically

significant difference". Better to make the comments in discussion. Addressed

5) Table 3 is repetitive. The categories are repeated several times. Addressed

6) in the discussion the statement that institutionalization correlates to falls is incorrect as there is no group of non-institutionalized elderly comparisons

 That statement is removed from the discussion.

Reviewer 02

Title: well done. Thankful for the comment.

Abstract: see methods’ comment, I would do the same changes about measures description. Abstract revised to incorporate the comment and to maintain the word limit.

Introduction: I would had that there have been different study investigating the correlation between different diet/ integration and the risk of falls (for example I would suggest to cite this study DOI: 10.1007/s40520-021-01977-x). The objective of the study did not include, exploring diet patterns and/or its association with risk of falls among older adults, therefore, information about diet patterns is not included

Methods

Measures: in this section I would be more specific describing which of the variable (bioelectrical impedance analysis, 2 minute walk test, 30 second sit to stand test, hand grip strength measure, chair sit and reach test, and back scratch test) is the measure of each outcome (Body composition, cardiovascular endurance, lower body strength and endurance, upper body strength, lower body flexibility, upper body flexibility). Addressed 

Results, discussion, and conclusion: well described. Thankful for the comment.

---

## [Editor Report · Decision Letter 1]

16 Jan 2024

Prevalence of falls and comparison of health-related physical fitness factors between different faller categories among institutionalized older adults in Kandy District of Sri Lanka

PONE-D-23-30015R1

Dear Dr. Suheja Madhushani Ihalage,

We’re pleased to inform you that your manuscript has been judged scientifically suitable for publication and will be formally accepted for publication once it meets all outstanding technical requirements.

Kind regards,

Antimo Moretti

Academic Editor

PLOS ONE

---

## [Editor Report · Acceptance letter]

10 Feb 2024

PONE-D-23-30015R1 

PLOS ONE

Dear Dr. Ihalage, 

I'm pleased to inform you that your manuscript has been deemed suitable for publication in PLOS ONE. Congratulations! Your manuscript is now being handed over to our production team.

Kind regards, 

on behalf of

Prof. Dr. Antimo Moretti 

Academic Editor

PLOS ONE